# Enhanced IL-17 Producing and Maintained Cytolytic Effector Functions of Gut Mucosal CD161^+^CD8^+^ T Cells in SIV-Infected Rhesus Macaques

**DOI:** 10.3390/v15091944

**Published:** 2023-09-18

**Authors:** Siva Thirugnanam, Edith M. Walker, Faith Schiro, Pyone P. Aye, Jay Rappaport, Namita Rout

**Affiliations:** 1Tulane National Primate Research Center, Covington, LA 70433, USA; tsivasakthivel@tulane.edu (S.T.); paye@tulane.edu (P.P.A.); jrappaport@tulane.edu (J.R.); 2Department of Microbiology and Immunology, Tulane University School of Medicine, New Orleans, LA 70112, USA; 3Tulane Center for Aging, Tulane University School of Medicine, New Orleans, LA 70112, USA

**Keywords:** mucosal immunity, T cells, CD161, SIV, IL-17, TNF-α

## Abstract

Previous studies have indicated that the loss of CD161-expressing CD4^+^ Th17 cells is linked to the progression of chronic HIV. These cells are significantly depleted in peripheral blood and gut mucosa of HIV-infected individuals, contributing to inflammation and disruption of the gut barrier. However, the impact of HIV infection on CD161-expressing CD8^+^ T cells remain unclear. Here, we examined the functions of peripheral blood and mucosal CD161^+^CD8^+^ T cells in the macaque model of HIV infection. In contrast to the significant loss of CD161^+^CD4^+^ T cells, CD161^+^CD8^+^ T cell frequencies were maintained in blood and gut during chronic SIV infection. Furthermore, gut CD161^+^CD8^+^ T cells displayed greater IL-17 production and maintained Th1-type and cytolytic functions, contrary to impaired IL-17 and granzyme-B production in CD161^+^CD4^+^ T cells of SIV-infected macaques. These results suggest that augmented Th17-type effector functions of CD161^+^CD8^+^ T cells during SIV infection is a likely mechanism to compensate for the sustained loss of gut mucosal Th17 cells. Targeting the cytokine and cytolytic effector functions of CD161^+^CD8^+^ T cells in the preclinical setting of chronic SIV infection with antiretroviral therapy has implications in the restoration of gut barrier disruption in persons with HIV infection.

## 1. Introduction

Chronic inflammation and disease progression in HIV infection is attributed to the disruption of the intestinal epithelial barrier along with the impairment of the mucosal immune response, leading to microbial translocation and opportunistic infections [1,2,3]. HIV and SIV infections in humans and macaques lead to early and substantial viral dissemination in the gut mucosa, resulting in severe CD4^+^ T cell depletion, predominantly the Th17 subsets [4]. Th17 cells critically contribute to mucosal defenses through their secretion of IL-17 and IL-22, which facilitate the production of antimicrobial molecules and strengthen the intestinal barrier by stimulating the proliferation of enterocytes and the transcription of tight-junction proteins, such as claudins [5,6,7,8]. Consequently, the substantial decrease in CD4^+^ Th17 cells during HIV/SIV infection is linked to the deterioration of mucosal barrier functions. This degradation leads to an increase in microbial translocation from the intestine to the bloodstream, ultimately contributing to persistent immune activation and disease progression [9,10,11,12]. Maintaining the functional abilities of Th17-type immune cells is therefore crucial for safeguarding the integrity of the gut barrier and decreasing the persistent immune activation associated with HIV infection.

A significant body of evidence suggests that other non-classical lymphocyte populations can act as an important source of mucosal IL-17, including innate-like T cells such as γδ T cells, MAIT cells, NKT cells, and various sub-populations of lineage negative, RORγt^+^ cells including ILCs [13]. Corresponding with the decline of Th17 cells, the IL-17/IL-22-producing subset of innate lymphoid cells, ILC3, has been observed to lose IL-17 cytokine functions and increase production of TNF-α, IFN-γ, and MIP-1β in SIV-infected animals [14,15,16]. While the impairment of the immune system and the reduction of Th17 cells in response to the loss of mucosal CD4^+^ T cells is well-documented in HIV/SIV infections [10,17,18], the impact on mucosal immune functions of other IL-17-producing T cell subsets remains partially understood [19].

We have previously reported that rhesus macaque CD161^+^ T cells represent mucosal tissue homing CD8 T cell subpopulations with Th1/Th17 type and cytotoxic effector functions [20]. CD161 is a C-type lectin-like receptor that belongs to the Killer cell lectin-like receptor subfamily (KLRB1), also known as NKR-P1. CD161 is expressed by a broad range of lymphocytes, including NK cells, CD4^+^, CD8^+^, γδT, NKT, and MAIT cells [21,22]. Human CD8^+^CD161^++^ T cells express a pattern of molecules associated with Th17 phenotype, including expression of RORγt, CCR6, and IL-18R along with cytokines IL-17, IL-22, and IFN-γ [23]. Additionally, human CD161-expressing CD8^+^ antiviral memory T cells exhibit polyfunctional and gut tissue-homing properties with enhanced effector functions, including cytokine production and high levels of cytotoxic mediators along with the expression of transcription factors T-bet and eomesodermin (EOMES), suggesting their eminent role in antiviral immunity and vaccine responses [24]. 

Our previous studies have shown that macaque CD161^+^ T cells are functionally analogous to human CD161^+^ T cells displaying Th1/Th17 type cytokine and cytotoxic effector functions, besides being readily available in mucosal tissues including lung and colon [20]. In this study, to further explore their role in gut mucosal immunity in the context of HIV infection, we examined the functions of non-invariant CD161^+^ T cell subpopulations in peripheral blood and mucosal tissues of rhesus macaques during chronic SIV infection. Our results highlight important differences between CD161-expressing CD8^+^ T cells and CD4^+^ T cells and suggest that CD161^+^CD8^+^ T cells may have important functions in gut mucosal immunity during chronic SIV/HIV infections.

## 2. Materials and Methods

### 2.1. Ethics Statement

The Tulane University Institutional Animal Care and Use Committee approved all procedures used during this study. The Tulane National Primate Research Center (TNPRC) is accredited by the Association for the Assessment and Accreditation of Laboratory Animal Care (AAALAC no. 000594). The U.S. National Institutes of Health (NIH) Office of Laboratory Animal Welfare number for TNPRC is A3071-01. 

### 2.2. Animals and Tissue Sampling

Biological specimens including blood, brochoalveolar lavage (BAL), and tissue samples were obtained from a total of 24 adult Indian rhesus macaques that were housed at TNPRC. Animals included healthy (negative for SIV, type D retrovirus, and STLV-1 infection) or chronic SIVmac251-infected macaques (18–26 weeks post-SIV infection). BAL and EDTA-blood and tissue specimens, including mesenteric lymph nodes, colon and jejunum segments, were obtained from uninfected or SIV-infected rhesus macaques undergoing euthanasia. 

### 2.3. Isolation of Lymphocytes

Blood samples collected in EDTA vacutainer tubes (Sarstedt Inc., Newton, NC, USA) were processed immediately. Peripheral blood mononuclear cells (PBMC) were separated by density gradient centrifugation (Lymphocyte Separation Medium; MP Biomedicals Inc., Solon, OH, USA) at 1500 rpm for 45 min and were used for phenotyping and in vitro functional assays. For the isolation of lymphocytes from colon and jejunum sections, a one-inch segment was collected in 50 mL tubes with ice-cold RPMI (Fisher Scientific, Waltham, MA, USA) and immediately transported to the laboratory. Fat and blood vessels were trimmed. Mucus and gross debris were removed, and the specimens were further washed twice in cold PBS (pH 7.4). Tissue samples were cut into small pieces (approximately 0.5–1 cm^2^) and processed as previously described [25]. In brief, tissues were treated with EDTA with shaking at 37 °C, followed by digestion in complete RPMI medium containing 5% fetal calf serum (FCS) (RPMI-5) containing 75 units/mL of Type II collagenase (Sigma-Aldrich, St. Louis, MI, USA) with shaking at 37 °C. For enrichment of lymphocytes, supernatants of LPLs were centrifuged over discontinuous Percoll (Sigma-Aldrich) density gradients followed by washing with PBS. All isolated cells were washed and re-suspended in complete RPMI-10 medium containing 10% FCS before staining or stimulation for functional assays. All cells were >90% viable by trypan blue dye exclusion method. Mesenteric lymph nodes were cut into small pieces (approximately 0.5–1 cm^2^) in a Petri-dish with cold PBS and squeezed with the back of a 10 mL syringe plunger to release the lymphocytes. The resulting suspension was filtered through 75 μm filter to obtain a uniform single-cell suspension of lymphocytes with >70% purity. BAL fluid (2 × 20 mL washes of sterile PBS) were obtained from macaque lungs and centrifuged at 950 g for 10 min at 4 °C to pellet cells and subsequently filtered through 100 μm filter, washed and re-suspended in complete RPMI-10 medium containing 10% FCS before fluorescent antibody staining or stimulation for functional assays.

### 2.4. Immunophenotyping Analysis

Multi-color flowcytometric analysis was performed on cells according to standard procedures using anti-human mAbs that cross-react with rhesus macaques as earlier described. Fresh or frozen cells were stained for flow cytometry as earlier described [19]. Briefly, 1–2 million cells, which were resuspended in 100 μL stain media (PBS with 2% FBS), were incubated with appropriate surface antibody cocktails for 30 min at 4 °C and washed with Running Buffer. For immunophenotyping analysis, PBMC were stained with the antibodies for CD45, CD3, CD4, CD8, CD161, γδ TCR, TCR Vα7.2, and PBS-57-loaded CD1d tetrmers (CD1dTM). For analysis of cytokine and chemokine receptor expression as trafficking markers, antibodies targeting CCR5, CCR6, CXCR3, CXCR5, IL-18Rα, CD69, and α_4_β_7_ were used. Antibody clones and sources are listed in Appendix A. Stained samples were resuspended in FACS Fixation and Stabilization Buffer (BD Biosciences, San Jose, CA, USA) and analyzed within 24 h after sample processing. Unstained samples were run with every set of samples. Data were acquired on a BD Fortessa with FACSDiva software. Analysis of the acquired data was performed using FlowJo software (version 10.9.0; TreeStar, Woodburn, OR, USA).

### 2.5. Functional Analysis

For flow cytometric evaluation of cytokine production and GranzymeB expression, 1–2 million PBMC or tissue lymphocytes were stimulated with Phorbol-12-Myristate-13-Acetate (PMA, 10 ng/mL, Sigma-Aldrich) and Ionomycin (1 μg/mL, Sigma-Aldrich) in the presence of brefeldin A (5 μg/mL, Sigma-Aldrich) for 12–16 h at 37 °C in 5% CO_2_. After activation, the cells were washed and stained appropriately as earlier described [26]. Briefly, mononuclear cells isolated from blood and tissues were stained extracellularly with CD3/CD161/CD8/CD4/γδTCR/Vα7.2TCR. The cells were then resuspended in Fixation/Permeabilization Buffer (BD Biosciences) for one hour, then washed twice with Perm/Wash Buffer (BD Bioscience). The intracellular antibodies for IFN-γ, IL-17, IL-22, TNF-α, and granzyme B were then added and incubated for 25 min at room temperature. Stained samples were processed as described under immunophenotyping analysis. 

### 2.6. Statistical Analyses

All statistical analysis was performed using GraphPad Prism Software (Version 10.0.1). Data were analyzed by using unpaired non-parametric Mann–Whitney test for comparisons between SIV-naïve and SIV-infected macaque groups. The Wilcoxon matched-pairs signed rank test was used to compute differences between the cell subpopulations within each group. *p* values of 0.05 or lower were considered significant, * *p* < 0.05, ** *p* < 0.01, *** *p* < 0.001, **** *p* < 0.0001.

## 3. Results

### 3.1. Impact of Chronic SIV Infection on Frequency and Phenotype of Circulating CD161^+^ T Cells in Rhesus Macaques

We have previously reported that CD161^+^ T cells represent mucosal tissue homing CD8 T cells with Th1/Th17-type cytokine production and cytotoxic potential [20], which are also referred to as Tc17 cells with antiviral properties [23,27]. By using CD161 expression on T cells as a surrogate marker for Th17 and Tc17 cells and employing the stringent flow cytometry gating strategy (Figure 1) that excludes CD161-expressing invariant T cell subpopulations, such as γδ T, MAIT, and iNKT cells, we first examined the impact of chronic SIV infection on CD161^+^ T cell subsets. 

Circulating frequencies evaluated in 13 SIV-naïve and 11 chronic SIV-infected rhesus macaques revealed significantly lower frequencies of total CD161^+^ T cells (*p* = 0.0018) in PBMCs from SIV-infected rhesus macaques when compared to SIV-naïve macaques (Figure 2). The chronic SIV infection period ranged from 18–26 weeks post-SIV inoculation, which is equivalent to early chronic HIV infection in humans without antiretroviral therapy. In concordance with earlier studies in HIV/SIV infected cohorts [17,18,28], lower frequencies of peripheral blood CD161^+^ T cells were most significant in chronic SIV-infected group of macaques for the CD4^+^ subsets (*p* < 0.0001), in comparison to the SIV-naïve group (Figure 2). Interestingly, however, there was no significant difference in CD161^+^CD8^+^ T cell frequencies between SIV-naïve and SIV-infected macaques (Figure 2). Together, these results suggest that a specific loss of CD161^+^CD4^+^ T cells mainly contribute the decline in peripheral blood CD161^+^ T cells during chronic SIV infection. 

Our earlier studies have shown that circulating CD161^+^CD8^+^ T cells in uninfected macaques expressed high levels of IL-18R and CD69 like their human equivalent, with nearly half of them expressing the gut homing integrin α4β7 and CXCR3 [20]. To determine changes in trafficking pattern during chronic SIV infection, we compared the phenotype and proportion of both CD161^+^CD8^+^ T and CD161^+^CD4^+^ T cells between SIV-naïve and SIV-infected macaques. We observed significantly greater levels of CD161^+^CD4^+^ T cells expressing CCR5 (*p* = 0.0031), and CXCR5 (*p* = 0.0069) in SIV-infected macaques, suggesting an increase in both Th1 type and secondary lymphoid organ-homing Th17-type subsets. However, no differences were observed in the phenotype and proportion of CD161^+^CD8^+^ T cells with respect to the expression of trafficking molecules or activation markers (Figure 3). Thus, the unaltered homing capacity and functional phenotype of the circulating CD161^+^CD8^+^ T cells during chronic SIV infection, despite the significant changes in the CD4^+^ subsets, supports their maintained functionality in the periphery to potentially act as antiviral immune cells. 

Within the SIV-naïve group, CD161^+^CD8^+^ T cells displayed a higher expression of the tissue retention marker CD69 and lower levels of CXCR3 in comparison to CD161^+^CD4^+^ T cells (Appendix A). On the other hand, within SIV-infected macaques, CD161^+^CD4^+^ T cells had significantly higher expression of CXCR5 and HLA-DR than CD161^+^CD8^+^ T cells (Appendix A) further supporting the enrichment of lymphoid tissue homing and activated CD161^+^ Th17-type cells that are reported to be preferred targets for HIV/SIV infection.

### 3.2. CD161^+^CD8^+^ T Cell Frequencies Are Maintained in Secondary Lymphoid and Mucosal Tissues during Chronic SIV Infection

Previous studies have shown that CD4 T cells expressing CD161 are depleted from blood during HIV and SIV infections [10,17,18] because of viral infection and redistribution to mucosal tissues along with significant loss of IL-17 functions [10]. However, the impact of HIV/SIV infection on tissue-resident CD161^+^CD8^+^ T cells remain unclear. To evaluate the impact of chronic SIV infection on CD161^+^CD8^+^ T cells in tissues, we next examined them in mesenteric lymph nodes, colon, jejunum, and BAL. No notable decrease in total CD161^+^ T cell frequencies were observed in lymph nodes and mucosal tissues of SIV-infected macaques when compared with SIV-naïve macaques (Figure 4A). Frequencies of CD161^+^CD4^+^ T cells were significantly lower in the lymph nodes and mucosal tissues of SIV-infected macaques (Figure 4B), confirming their continued depletion during chronic SIV infection in concordance with earlier studies [10]. In contrast, no significant differences were observed between the cohorts of SIV-infected and SIV-naïve macaques in CD161^+^CD8^+^ T cell frequencies in mesenteric lymph nodes, colon, jejunum, and BAL fluid (Figure 4C). Taken together with the data showing similar frequencies and trafficking phenotype of peripheral blood CD161^+^CD8^+^ T cells between macaques regardless of SIV infection status (Figure 2 and Figure 3), these data suggest that unlike CD161^+^CD4^+^ T cells, CD161^+^CD8^+^ T cells are maintained in secondary lymphoid and mucosal tissues during chronic SIV infection.

### 3.3. CD161^+^CD8^+^ T Cells Display Enhanced IL-17 Producing Function in Periphery and Gut Mucosa of SIV-Infected Macaques 

To investigate their role in tissue immunity during SIV infection, we examined the functions of CD161^+^ T cells in peripheral blood, mucosal tissues and lymph nodes of SIV-infected rhesus macaques. Among major cytokines produced by CD161-expressing T cells, including TNF-α, IFN-γ, IL-17, and IL-22, significant differences in IL-17 production were observed between CD161^+^CD4^+^ T and CD161^+^CD8^+^ T cells in the context of chronic SIV infection. In agreement with earlier studies [10], IL-17 production by peripheral blood CD161^+^CD4^+^ T cells was significantly lower in SIV-infected than SIV-naïve macaques (*p* = 0.0121; Figure 5A). Conversely, CD161^+^CD8^+^ T cells displayed greater IL-17 response to mitogen stimulation in SIV-infected macaques than that of SIV-naïve group (*p* = 0.026; Figure 5B), suggesting enhanced IL-17 functions during chronic SIV infection. The greater IL-17 producing function was specific to CD161-expressing CD8 T cells, since CD161-negative CD8 T cells did not display any notable difference between both groups (Figure 5C). CD161^+^ T cells expressed greater amounts of IL-17 than IL-22 (Figure 5A,B) and majority the of the cells were single cytokine producers in comparison to both the IL-17/IL-22-producing function (Appendix A). Notably, CD161^−^CD8^+^ T cells in the SIV-infected group displayed lower Th1 type cytokine production, including TNF-α (*p* = 0.0121) and IFN-γ (*p* = 0.0409), in comparison to SIV-naïve group of macaques.

Since CD161^+^ T cells are reported to be redistributed to mucosal tissues during chronic HIV/SIV infection [10,29] and display predominant Th1/Th17 functions, we next evaluated TNF-α, IFN-γ, IL-17, and IL-22 cytokine production in colonic lamina propria lymphocytes in our cohorts of SIV-naïve and SIV-infected macaques. Like peripheral blood, significantly lower frequencies of IL-17-producing CD161^+^CD4^+^ T cells were found in colonic mucosa of SIV-infected macaques (*p* = 0.0005; Figure 6A), indicating a loss in both numbers and Th17 cytokine function in colon during chronic SIV infection. Similarly, in agreement with the enhanced IL-17 functions in peripheral blood, colonic CD161^+^CD8^+^ T cells also displayed greater levels of IL-17 production in response to mitogen stimulation (*p* = 0.0005; Figure 6B). Further, a significantly higher level of IL-22 production was found in colonic CD161^+^CD8^+^ T cells of SIV-infected macaques (*p* = 0.0005; Figure 6B), suggesting an overall increase in Th17-type functions. No significant differences were observed in CD161^−^CD8^+^ T cell functions with regard to TNF-α, IFN-γ, IL-17, and IL-22 production in response to mitogen stimulation (Figure 6C). 

Since CD161-expressing T cells have a shared Th1/Th17 function [30], with a potential role in coordinating local tissue inflammation to better understand the impact of SIV infection on the overall functional response, we next assessed the ratios of IL-17/IFN-γ production in both subsets in the context of SIV infection. SIV-infected macaques displayed significantly lower IL-17/IFN-γ ratios in both peripheral blood (*p* = 0.0031; Figure 7A) and colonic CD161^+^CD4^+^ T cells (*p* = 0.0031; Figure 7B), but maintained them in CD161^+^CD8^+^ T cells (*p* = 0.0078; Figure 7A,B), while lower IL-17/IFN-γ ratios in CD161^+^CD4^+^ T cells were due to a significant loss in IL-17 production; CD161^+^CD8^+^ T cells maintained their ratios by increasing both IL-17 and IFN-γ cytokine producing functions, specifically in colon (Figure 6B).

### 3.4. Maintained Cytolytic Functions of Mucosal CD161^+^CD8^+^ T Cells in Airway and Gut Mucosa of Chronic SIV-Infected Macaques

In addition to Th1/Th17-type functions, primate CD161^+^CD8^+^ T cells and their murine equivalent NK1.1^+^CD8^+^ T cells exhibit prominent cytotoxic potential in tissues [20,29,31]. Therefore, we performed comparative analysis of Granzyme B production in mitogen stimulated CD161^+^ T cell subsets in blood, BAL, and colon lymphocytes from SIV-naïve and infected groups of macaques. Frequencies of cells expressing Granzyme B were significantly lower in CD161^+^CD4^+^ T cells of PBMC (*p* = 0.002), and colon (*p* = 0.0108), and showed a trend of reduction in BAL (*p* = 0.0519), suggesting the impaired cytolytic potential across immune compartments, including peripheral blood, gut, and airway (Figure 8A). Although CD161^+^CD8^+^ T cells also displayed impaired Granzyme B producing function in peripheral blood (*p* = 0.0008); notably, this function was not affected in colon or BAL cells (Figure 8B). CD161^−^CD8^+^ T cells for both SIV-naïve and SIV-infected groups of macaques showed a wide range in Granzyme B production and no significant difference in any of the tissue compartments examined (Figure 8C). Thus, these findings suggest that mucosal CD161^+^CD8^+^ T cells maintain cytolytic functions, despite significant and systemic impairment in CD161^+^CD4^+^ T cells during chronic SIV infection of rhesus macaques.

## 4. Discussion

Chronic immune activation and inflammation persist in HIV-infected individuals even with viral suppression during highly active anti-retroviral therapy (HAART). This is mainly driven by microbial translocation due to the decline of mucosal CD4^+^ T cells and the dysfunction of IL-17-producing T cells, which are essential for maintaining the intestinal epithelial barrier integrity [24,29]. A subset of IL-17-producing CD8^+^ T-cells characterized by CD161 expression (CD161^+^CD8^+^ T cells) has been shown previously to represent highly functional effector memory T cells in humans and rhesus macaques [20,24,29]. However, the impact of chronic HIV/SIV infection on this population and its relationship with CD161^+^CD4^+^ T cell impairment remains unclear. Our findings reveal that gut mucosal CD161^+^CD8^+^ T cells augment IL-17 production and maintain Th1-type and cytolytic functions during chronic SIV infection of rhesus macaques, despite significant impairment in IL-17/granzyme-B production in their CD4^+^ counterparts.

We have reported earlier that CD161^+^CD8^+^ T cells are enriched in mucosal tissues of rhesus macaques and are largely comprised of γδ T cells and TCR Vα7.2+ MAIT cells [20]. To distinguish classical CD161^+^CD8^+^ T cells from unconventional T cells, we employed a stringent gating strategy to exclude known unconventional T cell subpopulations such as NKT cells, γδ T cells and MAIT cells. This is critical since unconventional T cells are distinct in their function and antigen-specificity from classical T cells with a polyclonal T-cell receptor (TCR) repertoire, and display differences in kinetics of immune responses to infection [32]. To date, very few studies have examined CD161^+^CD8^+^ T cell functions by excluding MAIT cells or γδ T cells [33], and reports of systematic exclusion of total unconventional T cells are lacking. In this study, using the SIV-infected rhesus macaque model of HIV infection, we were able to compare the phenotype and functions of peripheral blood CD161^+^CD8^+^ T cells with those in mucosal tissues, as well as relate them to the classical CD161^+^CD4^+^ T cells.

Our results concur with previous studies demonstrating depletion of CD161-expressing CD4 T cells from blood during HIV and SIV infections [10,17,18], primarily due to viral infection and redistribution to mucosal tissues [10]. This was further supported by higher levels of CCR5 expression on CD161^+^CD4^+^ T cells in our cohort of SIV-infected macaques. It is interesting to note that despite the hallmark decline in CD161^+^CD4^+^ T cell numbers and IL-17 producing function during chronic SIV infection, our results demonstrated maintained frequencies and trafficking patterns of circulating CD161^+^CD8^+^ T cells in SIV-infected macaques, indicating an unaltered homing capacity and functional phenotype during chronic SIV infection, and thus implying their sustained availability in the periphery to potentially act as antiviral immune cells.

The most striking difference observed between CD161-expressing CD4^+^ and CD8^+^ T cell subsets in our study was in their frequencies in different immune compartments. While CD161^+^CD4^+^ T cells displayed widespread decline in blood and tissues including LN, airway (BAL), colon, and jejunum, CD161^+^CD8^+^ T cells maintained their proportions during chronic SIV infection. The general loss of CD161^+^CD4^+^ T cells supports the observations of activated Th17 cells as preferred targets of HIV/SIV infection [34,35]. Given that CD8 T cells also undergo immune exhaustion and lose the ability to maintain homeostatic proliferation and produce key cytokines during chronic HIV/SIV infection [36], our findings suggest that the CD161+ subset of CD8 T cells are likely more resistant to cytokine dysregulation and have better proliferative responses to maintain their numbers during chronic SIV infection. Indeed, CD161^+^CD8^+^ T cells have been shown to display greater proliferative potential than their CD161^−^ counterparts [37]. This is important considering the enhanced IL-17 functions of CD161^+^CD8^+^ T cells in chronic SIV-infected macaques in our study, suggesting that IL-17-producing CD161^+^CD8^+^ T cells may proliferate and expand their functions to compensate for the systemic loss of Th17 functions while maintaining their Th1-type effector potential.

The augmented Th17-type function was even more pronounced in colonic CD161^+^CD8^+^ T cells and included both IL-17 and IL-22 cytokine production in response to mitogen stimulation. This contrasts with an earlier study in HIV-infected individuals showing that activated CD161-expressing CD8^+^ T cells had a reduced capacity to produce IL-17 in comparison to that of healthy individuals, despite HAART [38]. This dysfunction of CD161^+^CD8^+^ T cells was associated with the persistent immune activation demonstrated by high expression of CD38 and programmed death 1 protein, and high levels of soluble CD14. This study examined IL-17-production in CD161-expressing CD8+ T cells, which can comprise significant proportions of γδ T cells and MAIT cells, as earlier reported [20]. As both these cell populations exhibit loss of IL-17 functions during chronic SIV and HIV infections [19,39,40,41], the loss of CD161+CD8+ T cell IL-17 functions in HIV-positive individuals reported in the earlier study may have included γδ T and MAIT cell dysfunction [38]. Moreover, based on our findings of enhanced Th17-type cytokine functions of CD161^+^CD8^+^ T cells during chronic SIV infection, we hypothesize that this difference may be attributed to the large range in the duration of infection in the HIV cohort that included patients up to 288 months post-diagnosis [38] which results in progressive immune exhaustion in the global T cell compartment despite HAART.

Although CD161^+^CD8^+^ T cells display both Th1- and Th17-type effector functions, enhanced IL-17/IFN-γ ratios were found in mucosal tissues, including colon and airways of SIV-naïve macaques [20] and genital mucosa of humans [29], suggesting an enrichment of epithelial barrier protective effector functions at relevant sites. In this regard, our study revealed maintained IL-17/IFN-γ ratios in CD161^+^CD8^+^ T cells in blood and colonic mucosa of SIV-infected macaques, further supporting their role in the immune response to persistent gut barrier disruption. Although, the enhanced IL-17/IL-22 producing function of CD161^+^CD8^+^ T cells is insufficient to prevent mucosal barrier impairment and microbial translocation and fully compensate for the loss of their CD4 counterparts during progressive SIV infection likely due to the continued viral replication driving a predominant inflammatory milieu and ongoing epithelial cell damage. Thus, it is possible that the ongoing barrier disruption overwhelms the repair process, which fails to catch up in the context of untreated HIV/SIV infection. We hypothesize that this immune function may be better maintained in the setting of ART-suppressed viral infection and may be further stimulated by in vivo immune modulation, such as cytokine therapy or specific agonist-induced in vivo activation.

Besides the enhanced IL-17/IL-22 producing ability and maintained TNF-α/IFN-γ cytokine production, mucosal CD161^+^CD8^+^ T cells also demonstrated similar Granzyme B production in both groups. Human and murine CD161^+^/NK1.1^+^ CD8^+^ T cells are known to exhibit elevated cytotoxic potential upon activation, with enhanced expression levels of granzyme, perforin, and innate-like receptors in comparison to CD161^−^/NK1.1^−^ counterparts [31,42,43]. Thus, combined with the mucosal tissue homing properties, cytolytic CD161^+^CD8^+^ T cells may play an important role at sites of HIV/SIV entry and replication [24]. Our results, showing lower circulating frequencies of Granzyme B-production, suggest that cytolytic CD161^+^CD8^+^ T cells are likely trafficking in response to continued viral replication in mucosal tissues during chronic SIV infection. Potentially, distinct subpopulations of CD161^+^CD8^+^ T cells are involved in gut repair and cytolytic functions. Future studies with scRNAseq may identify these subsets and reveal targets for induction of specific functions in these subsets that are known to replicate efficiently and respond to pharmacological stimulation [31]. Indeed, in vitro treatment with anti-inflammatory agent sulfasalazine was shown to increase IL-17 production by CD161-expressing CD8^+^ T cells from HIV-infected patients [38].

The main strength of this study is in the stringent definition for CD161^+^CD8^+^ T cells by excluding the three distinct subpopulations of unconventional T cells that can also express CD161 and CD8 cell surface receptors, thus, enabling to rigorously address classical CD161^+^CD8^+^ T cell functions in blood and tissues during SIV infection. One of the limitations of our study, however, is the lack of longitudinal data on gut barrier functions and viral loads in relation to CD161^+^CD8^+^ T cell functions, which makes it challenging to determine a direct role in epithelial barrier functions during SIV infection. Nevertheless, our cross-sectional comparisons revealed significant differences between CD161-expressing CD4^+^ and CD8^+^ T cell subsets in the context of chronic SIV infection, underscoring the potential cytolytic and non-cytolytic protective functions of these cells in blood and mucosal tissue compartments. Future studies focusing on the signaling events inducing effector functions downstream of CD161 activation and the role of costimulatory and signaling molecules during CD161 engagement are needed to delineate their role in antiviral functions during chronic HIV/SIV infections.

In summary, we show that CD161^+^CD8^+^ T cell frequencies and function are unaffected during chronic SIV infection in rhesus macaques. Moreover, gut-residing CD161^+^CD8^+^ T cells displayed an enhanced IL-17 producing ability and sustained their Th1-type and cytolytic functions. Overall, our results reveal that future studies are needed in the pre-clinical setting of antiretroviral drug-treated SIV infection to determine the antiviral functions of gut mucosal CD161^+^CD8^+^ T cells and their role in regulation of tissue-specific inflammation and virus clearance.

## Figures and Tables

**Figure 1 viruses-15-01944-f001:**
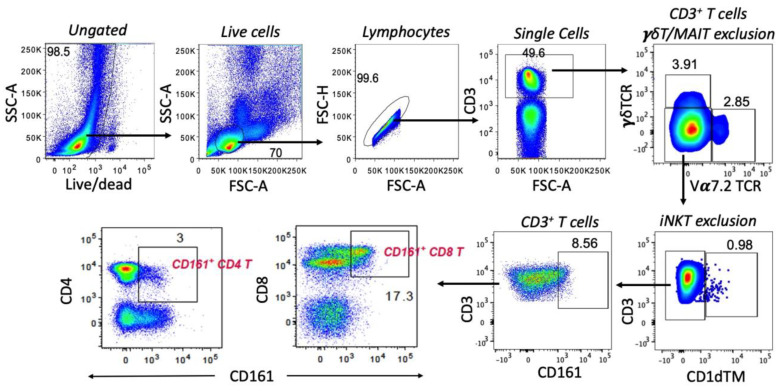
Representative flow cytometry plots of lymphocytes in PBMC showing gating strategy used for identification of CD8^+^ and CD4^+^ CD161^+^ T cells. Staining for γ/δ T cell receptor (γδTCR) and T cell receptor Vα7.2 on CD3^+^ T lymphocytes was used to exclude γδ T and MAIT, cells. On the γδ/Vα7.2TCR double negative T cells, staining for PBS–57–loaded CD1dTMs was used to exclude invariant NKT (iNKT) cells.

**Figure 2 viruses-15-01944-f002:**
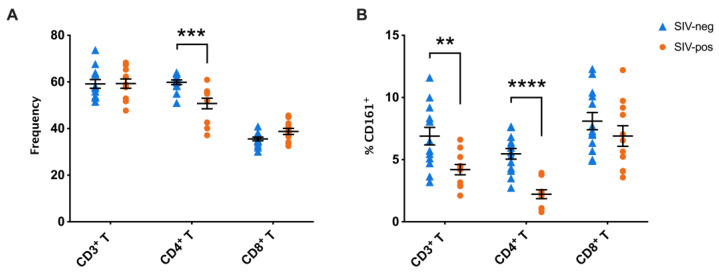
Maintained CD161^+^CD8^+^ T cells in peripheral blood despite highly significant decline in CD161^+^CD4^+^ T cells in chronic SIV-infected rhesus macaques. Frequencies of total CD3^+^ T, CD4^+^ T, and CD8^+^ T cells (**A**) and CD3^+^CD161^+^ T, CD161^+^CD4^+^ T, and CD161^+^CD8^+^ T cell subpopulations (**B**) are shown for PBMC isolated from 13 SIV-negative macaques (blue triangles) and 11 chronic SIV-infected macaques (orange circles). Error bars show mean and SEM for each group. Groups were compared using nonparametric Mann–Whitney test. Asterisks indicate significant differences between the groups (** *p* < 0.01; *** *p* < 0.001; **** *p* < 0.0001).

**Figure 3 viruses-15-01944-f003:**
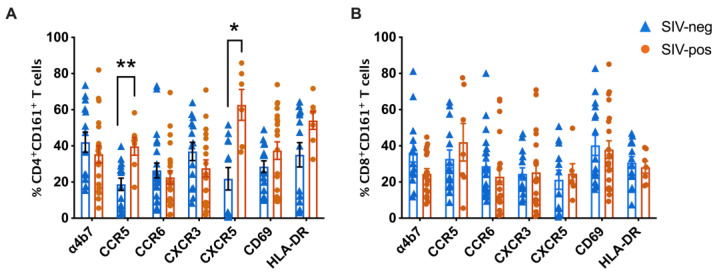
Higher levels of CCR5 and CXCR5 expressing CD161^+^CD4^+^ T cells and unaltered trafficking pattern of CD161^+^CD8^+^ T cells in peripheral blood of chronic SIV-infected rhesus macaques. Frequencies of trafficking markers (α4β7, CCR5, CCR6, CXCR3, and CXCR5) and activation markers (CD69 and HLA-DR) on (**A**) CD161^+^CD4^+^ T cells, and (**B**) CD161^+^CD8^+^ T cells are shown for PBMC isolated from 13 SIV-negative macaques (blue triangles) and 11 chronic SIV-infected macaques (orange circles). Error bars show mean and SEM for each group. Groups were compared using nonparametric Mann–Whitney test. Asterisks indicate significant differences between the groups (* *p* < 0.05; ** *p* < 0.01).

**Figure 4 viruses-15-01944-f004:**
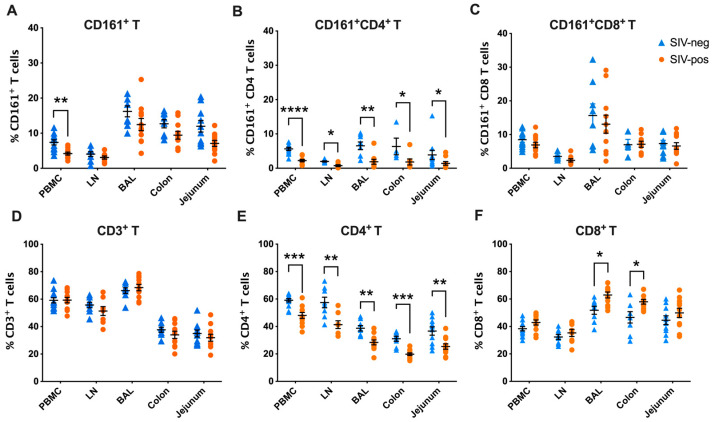
Frequencies of CD161^+^ T cell subsets in blood and tissues of SIV-negative and chronic SIV-infected rhesus macaques. Comparison of frequencies of CD3^+^CD161^+^ T, CD161^+^CD4^+^ T, and CD161^+^CD8^+^ T cell subpopulations (**A**–**C**) and total CD3^+^ T, CD4^+^ T, and CD8^+^ T cells (**D**–**F**) in PBMC, mesenteric lymph nodes (LN), BAL, colon and jejunum of SIV-negative macaques (n = 13, blue triangles) and chronic SIV-infected macaques (n = 11, orange circles). Error bars show mean and SEM for each group. Groups were compared using nonparametric Mann–Whitney test. Asterisks indicate significant differences between the groups (* *p* < 0.05; ** *p* < 0.01; **** *p* < 0.0001).

**Figure 5 viruses-15-01944-f005:**
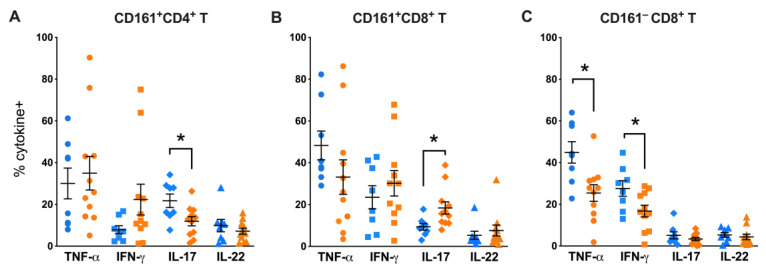
Greater IL-17 producing ability and maintained Th1-type cytokine production in circulating CD161^+^CD8^+^ T cells during chronic SIV infection. Intracellular cytokine staining of CD161^+^CD4^+^ T cells (**A**), CD161^+^CD8^+^ T cells (**B**), and CD161^−^CD8^+^ T cells (**C**) using antibodies for TNF-α, IFN-γ, IL-17, and IL-22 in PMA/Ionomycin stimulated PBMC from SIV-negative (blue symbols) and SIV-positive (orange symbols) macaques. Flow cytometry analysis demonstrated significantly lower frequency of IL-17^+^ CD161^+^CD4^+^ T cells and higher frequency of IL-17^+^ CD161^+^CD8^+^ T cells in the SIV-positive group. CD161^−^CD8^+^ T cells were examined to compare with CD161^+^CD8^+^ T cell functions and showed no increase in IL-17, but lower TNF-α and IFN-γ in the SIV-positive group compared to SIV-negative group of macaques. Error bars show mean and SEM for individual cytokines within each group. Groups were compared using nonparametric Mann–Whitney test. Asterisks indicate significant differences between the groups (* *p* < 0.05).

**Figure 6 viruses-15-01944-f006:**
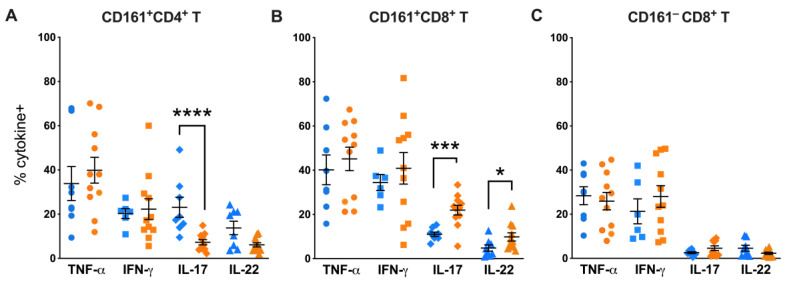
Enrichment of Th17-type effector functions in colonic mucosal CD161^+^CD8^+^ T cells during chronic SIV infection. Intracellular cytokine staining of CD161^+^CD4^+^ T cells (**A**), CD161^+^CD8^+^ T cells (**B**), and CD161^−^CD8^+^ T cells (**C**) for TNF-α, IFN-γ, IL-17, and IL-22 in PMA/Ionomycin stimulated lamina propria lymphocytes isolated from colon tissue obtained from SIV-negative (blue symbols) and SIV-positive (orange symbols) macaques. Flow cytometry analysis demonstrated significantly lower frequency of IL-17^+^ CD161^+^CD4^+^ T cells (*p* = 0.0005) and higher frequency of IL-17^+^ CD161^+^CD8^+^ T cells (*p* = 0.0025), and IL-22^+^ CD161^+^CD8^+^ T cells (*p* = 0.04) in the SIV-positive group. Error bars show mean and SEM for individual cytokines within each group. Groups were compared using nonparametric Mann–Whitney test. Asterisks indicate significant differences between the groups (* *p* < 0.05, *** *p* < 0.001; **** *p* < 0.0001).

**Figure 7 viruses-15-01944-f007:**
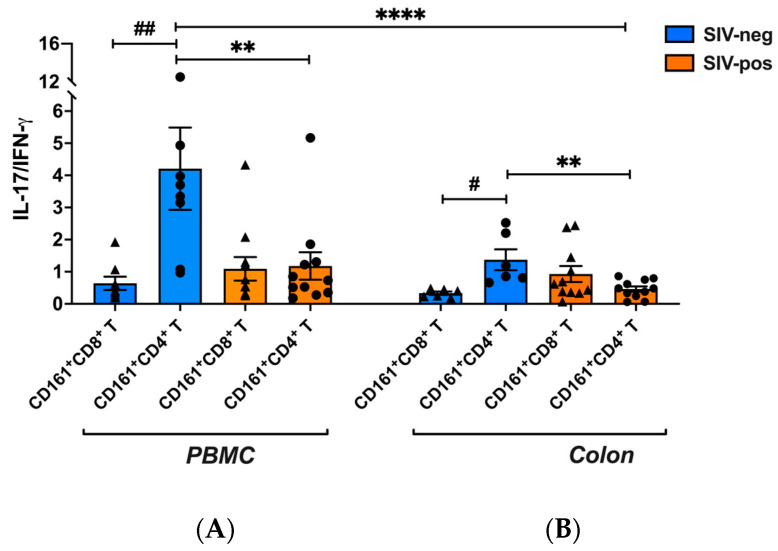
IL-17/IFN-γ ratios are reduced in peripheral blood and colonic CD161^+^CD4^+^ T cells but not in CD161^+^CD8^+^ T cells during chronic SIV infection. Ratios of IL-17 to IFN-γ producing cells in CD161^+^CD4^+^ T cells and CD161^+^CD8^+^ T cells in PBMC and colon of SIV-negative (blue bars) and SIV-infected (orange bars). Differences within group between the two cell subpopulations were calculated by the Wilcoxon matched-pairs signed rank test, while the SIV-negative and infected groups were compared using Wilcoxon matched-pairs signed rank test evaluated using nonparametric Mann–Whitney test. Error bars show mean and SEM. Hash indicates significant differences within each group (# *p* < 0.05; ## *p* < 0.01). Asterisks indicate significant differences between the groups by Mann–Whitney test (** *p* < 0.01; **** *p* < 0.0001).

**Figure 8 viruses-15-01944-f008:**
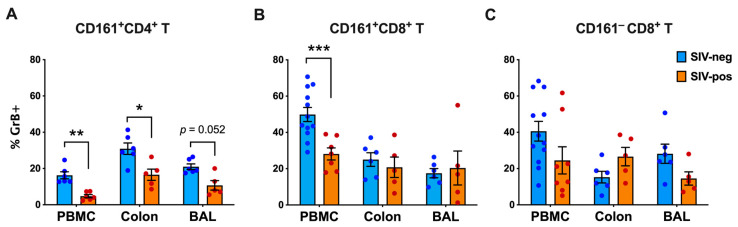
Impact of chronic SIV infection on intracellular Granzyme B production by CD161^+^CD4^+^ T, CD161^+^CD8^+^ T, and CD161^−^CD8^+^ T cells in peripheral blood and mucosal tissues. Intracellular GrB staining of CD161^+^CD4^+^ T cells (**A**), CD161^+^CD8^+^ T cells (**B**), and CD161^−^CD8^+^ T cells (**C**) in PMA/Ionomycin stimulated PBMC, BAL, and colon lamina propria lymphocytes of SIV-negative (blue bars) and SIV-positive macaques (orange bars). Significantly lower frequency of GrB^+^ cells were observed in CD161^+^CD4^+^ T cells in PBMC and colon. CD161^+^CD8^+^ T cells displayed lower GrB^+^ frequency in PBMC but not in colon or BAL. CD161^−^CD8^+^ T cells showed no significant difference between the groups in any of the 3 compartments assessed. Groups were compared using nonparametric Mann–Whitney test. Error bars show mean and SEM. Asterisks indicate significant differences between the groups (* *p* < 0.05, ** *p* < 0.01; *** *p* < 0.001).

## Data Availability

All the data obtained during this study are included in the manuscript. Additional information could be provided by the authors upon a reasonable request.

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
