# Peer review of "Enhanced IL-17 Producing and Maintained Cytolytic Effector Functions of Gut Mucosal CD161+CD8+ T Cells in SIV-Infected Rhesus Macaques"

_viruses, 2023, doi:10.3390/v15091944_

Round 1

Reviewer 1 Report

This manuscript describes the maintenance of CD161+ CD8 T cell population and their IL-17 and GzB expression in chronically SIV-infected rhesus macaques. This is in contrast to the significant loss or decrease of CD161+ CD4 T cells and their IL-17 and GrB expression during chronic SIV infection. The strength of the manuscript comes from the highly selective gating of the cell populations being studied, in which the gamma/delta T, MAIT, and iNKT cells were excluded prior to the selection of CD161+ CD8 and CD4 T cells. The study also compared changes of these T cell populations in the peripheral blood as well as in the mucosa (gut and BAL). The findings are of interest because they offer a possibility that harnessing these CD8 T cells in lieu of their CD4 counterparts may improve the integrity of mucosal barrier that has been implicated in persistent immune hyper-activation during HIV/SIV infection.

The authors should address the following points to improve the manuscripts:

1)      The author should comment as to why the increase of these IL-17+ CD161+ CD8 T cells in the gut mucosa is still insufficient to fully compensate the loss of their CD4 counterparts, as mucosal barrier impairment and microbial translocation still occur during infection. The authors may also discuss ideas about what it takes for the IL-17+ CD161+ CD8 T cells to restore or preserve the gut mucosal integrity and prevent chronic immune activation.

2)      Line 61: Add “Human” to specify “Human CD8+CD161++ T cells express a pattern of molecules… [23]”.

3)      Figure 1: In gating CD161+ T cells, the MFI cut-off for CD4+ and CD8+ T cells are not the same.  The CD161 positive gating of CD3+ T cells looks consistent with that of CD8+ T cells, but not with CD4+ T cells.

4)      The detection of cytokine and GzB expression is based on stimulation with PMA/ionomycin only. While PMA/ionomycin is a potent stimulus for activating potentially all cell populations, no experiments were conducted to test the responses to more physiologic antigens or mitogens.

5)      Certain subsets of CD161+ T cells are likely to express IL-17 and IL-22, along with GzB and Th1-type cytokines. The authors analyzed their expression singly. It would be of interest to evaluate the poly-functionality of the CD161+ CD8 T cells in comparison to the CD4 T cell counterparts and determine which polyfunctional or monofunctional subsets were altered most significantly during SIV infection. For example, are most of the GzB+ CD8+ and GzB+ CD4+ T cells also expressing IL-17 or IL-22 or both? How do they change with SIV infection?

6)      Line 319: Remove “To investigate”

7)      Animals were infected with SIVmac251 for >4 months. The range of infection period should be specified. What is the equivalent period for HIV infection in humans? Do the author expect the percentages of CD161+ CD8 and CD4 T cells to be changing over time during SIV infection? Do the author expect the observed findings to be specific for SIVmac251 or may they be applicable to other SIV strains?

Author Response

We thank the reviewer for their constructive feedback! Following are the point-by-point responses to the review comments:

  • The author should comment as to why the increase of these IL-17+ CD161+ CD8 T cells in the gut mucosa is still insufficient to fully compensate the loss of their CD4 counterparts, as mucosal barrier impairment and microbial translocation still occur during infection. The authors may also discuss ideas about what it takes for the IL-17+ CD161+ CD8 T cells to restore or preserve the gut mucosal integrity and prevent chronic immune activation.

We agree with this point that enhanced IL-17 functions of CD161+ CD8 T cells is still insufficient to fully compensate the loss of their CD4 counterparts during SIV infection. We speculate that the reason behind persistence of mucosal barrier impairment and microbial translocation during infection despite the increase in gut mucosal IL-17+ CD161+ CD8 T cells is because of continued viral replication in blood and gut during chronic SIV infection driving a predominant inflammatory milieu with ongoing epithelial cell damage. Therefore, the ongoing barrier disruption is likely overwhelming the repair process, which fails to catch up. Based on the reviewer’s sensible suggestion, we have now added this explanation to our discussion (lines 434–442). We speculate that this immune function may be better maintained in the setting of ART-suppressed viral infection and may be further stimulated by in vivo immune modulation such as cytokine therapy or specific agonist-induced activation.

  • Line 61: Add “Human” to specify “Human CD8+CD161++ T cells express a pattern of molecules… [23]”.

Made this edit.

  • Figure 1: In gating CD161+ T cells, the MFI cut-off for CD4+ and CD8+ T cells are not the same. The CD161 positive gating of CD3+ T cells looks consistent with that of CD8+ T cells, but not with CD4+ T cells.

It is established in ours and other studies that CD161 MFI is greater in CD8 T cells than CD4 T cells (PMID: 25437561; Fig. 5B). Indeed, CD8 T cells have 3 MFI levels distinguishing them as CD161-low, CD161-moderate and CD161-hi cells, with the CD161hi serving as a surrogate marker for MAIT cells (PMID: 26124758, PMID: 27309719, PMID: 26220166).

  • The detection of cytokine and GzB expression is based on stimulation with PMA/ionomycin only. While PMA/ionomycin is a potent stimulus for activating potentially all cell populations, no experiments were conducted to test the responses to more physiologic antigens or mitogens.

We agree that responses to more physiologic antigens would improve the understanding of antigen-specific responses of CD161+CD8+ T cells during SIV infection. The objective of this study, however, was to evaluate the overall functional phenotype and effector responses of CD161+CD8+ T cells in SIV-naive and SIV-infected animals in order to understand general impact of HIV infection on immune functions of this subpopulation in humans. As the reviewer pointed put, these results lay the foundation for our ongoing studies to further evaluate CD161+CD8+ T cell responses to more physiologic antigens in SIV-infected macaques with or without ART suppression of viremia.

  • Certain subsets of CD161+ T cells are likely to express IL-17 and IL-22, along with GzB and Th1-type cytokines. The authors analyzed their expression singly. It would be of interest to evaluate the poly-functionality of the CD161+ CD8 T cells in comparison to the CD4 T cell counterparts and determine which polyfunctional or monofunctional subsets were altered most significantly during SIV infection. For example, are most of the GzB+ CD8+ and GzB+ CD4+ T cells also expressing IL-17 or IL-22 or both? How do they change with SIV infection?

Again, an important question to further investigate in a longitudinal study. Most of the samples used in this cross-sectional study were from archived samples from previous studies and all the cytokines used were not in the same panel/sample tube precluding polyfunctionality analyses. In a new supplementary figure (#2), we have now included representative flow cytometry data on CD161+ T cells expressing combinations of IL-17 and/or IL-22 in a SIV-infected animal where the cytokines were present in the same panel.

  • Line 319: Remove “To investigate”

Thank you for pointing this out, we have removed it.

7)      Animals were infected with SIVmac251 for >4 months. The range of infection period should be specified. What is the equivalent period for HIV infection in humans? Do the author expect the percentages of CD161+ CD8 and CD4 T cells to be changing over time during SIV infection? Do the author expect the observed findings to be specific for SIVmac251 or may they be applicable to other SIV strains?

The infection period ranged from 18-26 weeks post-SIV inoculation. This is equivalent to early chronic HIV infection without antiretroviral therapy in humans. We have added this information to materials and methods section (line #90) and in the results section (line# 177-178).

Based on earlier report by MaGary et al., Mucosal Immunol 2017 (PMID: 28051083), the total CD161+ CD8+ T cells (cross-sectional comparison of SIV-naïve with ~ 1yr SIV-infected rhesus macaques) and CD161+ CD4+ T cells (longitudinal comparison of baseline to up to 23 weeks post-SIV) are lower during chronic SIV infection. Thus, we anticipate CD161+ CD4 T cells to decline during progressive SIV/HIV infection due to selective loss as well as overall CD4 T cell decline. However, as we pointed out in our study, the loss of CD161+ CD8+ T cells observed in other studies may be contributed by loss of MAIT cells and gamma delta T cells. As also suggested by McGary et al, that the observed lower CD161+ CD8+ T cell frequencies in cross-sectional comparison of SIV-naïve with ~ 1yr SIV-infected rhesus macaques represents the well-documented loss of mucosal associated invariant T (MAIT) cells in HIV-infected individuals.

Lastly, we anticipate our findings to be applicable to other strains of SIV as well as HIV infection, since they are not targets of direct SIV/HIV infection and are likely activated by nonclassical (non-peptide) antigens that are conserved across strains.

Reviewer 2 Report

The manuscript describes a potential role of CD161+ CD8+ T cells in participating immunity in HIV+ patients. The MS is sound and describes the potential role of, often neglected, CD8 T cells. Although the immunity role is not clear, i.e., progressive vs. chronic, this study illuminates the particular role of CD8 T cells in sustaining immunity, perhaps compensating the CD4 T cels at larger context. However, addressing following comments help understand the role of CD161+CD8+ T cells in a nuance way.

Major:

1. The MS needs the frequency of CD8 and CD4 T cells among the T cells in all relevant figures data, esp. Fig.2 and Fig.4.

2. can authors show CD161+CD8+ T cells expressing IL-17 and IL-22 dual producing cells?

Minor: 

1. CD1dTM is mentioned but not able to get the MS data out of it clearly.

2. Line 315. Fig. number is rightly put? 

3. Typos throughout the MS needs to be checked. 

Overall English is good and acceptable. May need to look into some typos.

Author Response

Thank you for the helpful comments. Following are the point-by-point responses:

  • The MS needs the frequency of CD8 and CD4 T cells among the T cells in all relevant figures data, esp. Fig.2 and Fig.4.

We have now included CD3, CD4, and CD8 T cell frequencies as Fig.2B and Fig.4 (D-F).

  • Can authors show CD161+CD8+ T cells expressing IL-17 and IL-22 dual producing cells?

We have previously shown IL-17/IL-22-producing polyfunctional CD161+CD8+ T cells T cells in a longitudinal study of SIV infected macaques (Supplementary figure 4 in Walker et al., Frontiers in Immunology, 2021).

As the samples used in the cross-sectional study presented in this manuscript were from archived samples from previous studies and all the cytokines used were not in the same panel/sample tube precluding polyfunctionality analyses. In a supplementary figure, we have now included representative flow cytometry data on CD161+ T cells expressing combinations of IL-17 and/or IL-22 in a SIV-infected animal where the cytokines were present in the same panel.

  • CD1dTM is mentioned but not able to get the MS data out of it clearly.

CD1DTM was used to exclude iNKT cells from the gating for CD161+ T cell subsets. It is shown in the gating strategy in Figure 1. The manuscript data does not include iNKT data.

  • Line 315. Fig. number is rightly put?

Yes, we are referring to the earlier figure 6B that shows the data as individual cytokines. “While lower IL-17/IFN-gamma ratios in CD161+CD4+ T cells was due to significant loss in IL-17 production, CD161+CD8+ T cells maintained their ratios by increasing both IL-17 and IFN- gamma cytokine producing functions, specifically in colon (Figure 6B).”

  • Typos throughout the MS needs to be checked.

We have corrected one typographical error pointed out by other reviewer. Could the reviewer kindly point us to the typos elsewhere if we have missed additional ones.
